

# High-resolution inversion of methane emissions in the Southeast US using SEAC[4]RS aircraft observations of atmospheric methane: anthropogenic and wetland sources

Jian-Xiong Sheng[1], Daniel J. Jacob[1], Alexander J. Turner[1,5], Joannes D. Maasakkers[1], Melissa P. Sulprizio[1], A. Anthony Bloom[2], Arlyn E. Andrews[3], and Debra Wunch[4]

[1]School of Engineering and Applied Sciences, Harvard University, Cambridge, MA, USA
[2]Jet Propulsion Laboratory, California Institute of Technology, Pasadena, CA, USA
[3]NOAA Earth System Research Laboratory, Boulder, Colorado, USA
[4]Department of Physics, University of Toronto, Toronto, Canada
[5]now at Department of Earth and Planetary Sciences, University of California at Berkeley, CA, USA

*Correspondence to:* Jian-Xiong Sheng (jsheng@seas.harvard.edu)

**Abstract.** We use observations of boundary layer methane from the SEAC[4]RS aircraft campaign over the Southeast US in August-September 2013 to estimate methane emissions in that region through an inverse analysis with up to $0.25° \times 0.3125°$ ($25 \times 25$ km$^2$) resolution and with full error characterization. The Southeast US accounts for about half of total US anthropogenic emissions according to the gridded EPA national inventory and also has extensive wetlands. Our inversion uses state-of-science emission inventories as prior estimates, including a gridded version of the anthropogenic EPA Greenhouse Gas Inventory and the mean of the WetCHARTs ensemble for wetlands. Inversion results are independently verified by comparison with surface (NOAA/ESRL) and column (TCCON) methane observations. Our posterior estimates for the Southeast US are $12.8\pm0.9$ Tg a$^{-1}$ for anthropogenic sources (no significant change from the gridded EPA inventory) and $9.4\pm0.8$ Tg a$^{-1}$ for wetlands (27% decrease from the mean in the WetCHARTs ensemble). The largest source of error in the WetCHARTs wetlands ensemble is the landcover map specification of wetland areal extent. We find no regional bias in the anthropogenic EPA inventory, including for different source sectors, in contrast with previous inverse analyses that found the EPA inventory to be too low at national scales. These previous inversions relied on prior anthropogenic source patterns from the EDGAR v4.2 inventory that have considerable error, and also assumed low wetland emissions. Despite the regional-scale consistency, we find significant local errors in the EPA inventory for oil/gas production fields, suggesting that emission factors are more variable than assumed in the inventory.



## 1   Introduction

Methane is an important greenhouse gas (Myhre et al., 2013) for which individual countries report national emissions to the United Nations Framework Convention on Climate Change (UNFCCC; United Nation, 1992). Observations of atmospheric methane reviewed by Brandt et al. (2014) have implied that the US national inventory reported by the Environmental Protection

Agency (EPA) may be greatly underestimated. Here we use aircraft observations from the NASA SEAC[4]RS aircraft campaign over the Southeast US (Toon et al., 2016), together with a newly gridded version of the EPA inventory (Maasakkers et al., 2016), in a fine-resolution inversion with detailed error characterization to better quantify the sources of methane emissions over this major source region.

The EPA (2016) reports a national anthropogenic emission total of 29.2 Tg a$^{-1}$ for 2014, with no significant trend over the

past decade and less than $\pm 3\%$ interannual variability. Major contributors are livestock (32%) , the oil/gas industry (32%), waste (22%), and coal mining (8%). The EPA (2016) inventory is consistent with Lyon et al. (2015) for oil/gas systems and Wolf et al. (2017) for livestock, and 8% higher than the previous versions (EPA, 2013, 2014), largely due to updated oil/gas emissions. There is also a highly uncertain natural source from wetlands, estimated at 4.5-14 Tg a$^{-1}$ for the contiguous US in the WETCHIMP compilation of inventories (Melton et al., 2013). Inverse analyses of atmospheric methane observations have

suggested that the EPA bottom-up inventory (EPA, 2013, 2014) is too low by about 30% (Miller et al., 2013; Turner et al., 2015; Alexe et al., 2015), but they relied on prior estimates from the global EDGAR v4.2 inventory (European Commission, 2011) that have large errors in source patterns particularly for oil/gas systems (Maasakkers et al., 2016; Sheng et al., 2017). For example, EDGAR v4.2 does not account for the large source from oil/gas production in the Southeast US but the gridded EPA inventory does (Maasakkers et al., 2016). Errors in source patterns used as prior estimates can greatly bias inversion results

(Jacob et al., 2016).

The SEAC[4]RS aircraft campaign conducted in August-September 2013 offers an opportunity for better estimating methane emissions in the Southeast US, a region that accounts for about half of anthropogenic methane emissions in the US according to the gridded EPA inventory (Maasakkers et al., 2016) and also has extensive wetlands. The aircraft flights provided extensive boundary-layer measurements of methane across the region. We conduct an inverse analysis of the SEAC[4]RS data with the

GEOS-Chem chemical transport model (CTM) at $0.25° \times 0.3125°$ resolution, using state-of-science prior estimates from the gridded EPA inventory of Maasakkers et al. (2016) and the WetCHARTs extended ensemble wetlands inventory of Bloom et al. (2017). This allows us to evaluate the EPA inventory with better accuracy and resolution than has been done before, and also to gain better understanding of US wetland emissions.

## 2   Methods

We derive an optimized estimate of spatially resolved methane emissions in the Southeast US (domain of Fig. 1) by Bayesian inverse analysis of atmospheric methane observations from the SEAC[4]RS aircraft campaign. Let the vector **x** represent a gridded ensemble of methane emissions in the region (state vector for the inversion).The inversion minimizes the cost function





$J(\mathbf{x})$ by solving $\nabla_{\mathbf{x}} J(\mathbf{x}) = \mathbf{0}$:

$$J(\mathbf{x}) = (\mathbf{x} - \mathbf{x_A})^T \mathbf{S_A^{-1}}(\mathbf{x} - \mathbf{x_A}) + (\mathbf{y} - \mathbf{Kx})^T \mathbf{S_O^{-1}}(\mathbf{y} - \mathbf{Kx}). \quad (1)$$

Here the methane observations are assembled as a vector $\mathbf{y}$, $\mathbf{x_A}$ is the prior emission estimate, $\mathbf{K}$ is the Jacobian matrix describing the sensitivity of concentrations to emissions, and $\mathbf{S_A}$ and $\mathbf{S_O}$ are the prior and observational error covariance

matrices, respectively.

Analytical solution of $\nabla_{\mathbf{x}} J(\mathbf{x}) = \mathbf{0}$ yields the optimal estimate $\hat{\mathbf{x}}$, the posterior error covariance matrix $\hat{\mathbf{S}}$, and the associated averaging kernel matrix $\mathbf{A}$ (Rodgers, 2000; Brasseur and Jacob, 2017)

$$\hat{\mathbf{x}} = \mathbf{x_A} + \mathbf{S_A} \mathbf{K}^T (\mathbf{K} \mathbf{S_A} \mathbf{K}^T + \mathbf{S_O})^{-1}(\mathbf{y} - \mathbf{Kx_A}), \quad (2)$$

$$\hat{\mathbf{S}}^{-1} = \mathbf{K}^T \mathbf{S_O^{-1}} \mathbf{K} + \mathbf{S_A^{-1}}, \quad (3)$$

$$\mathbf{A} = \mathbf{I}_n - \hat{\mathbf{S}} \mathbf{S_A^{-1}}. \quad (4)$$

where $\mathbf{I}_n$ is the identity matrix with $n$ being the dimension of the state vector $\mathbf{x}$. Inversions of atmospheric methane observations usually solve $\nabla_{\mathbf{x}} J(\mathbf{x}) = \mathbf{0}$ numerically using an adjoint method (Henze et al., 2007). The analytical solution has the advantage

of providing complete error characterization of the optimal estimate $\hat{\mathbf{x}}$ through its error covariance matrix $\hat{\mathbf{S}}$. The related averaging kernel matrix $\mathbf{A}$ describes the sensitivity of the optimal estimate $\hat{\mathbf{x}}$ to the true emissions $\mathbf{x}$. The trace of $\mathbf{A}$ quantifies the Degrees of Freedom For Signal (DOFS), i.e., the number of pieces of information in the observing system for constraining the methane emissions (DOFS $\leq n$).

The Jacobian matrix $\mathbf{K}$ for the inversion is constructed with the GEOS-Chem CTM (http://www.geos-chem.org), which

relates methane emissions to atmospheric concentrations through simulation of atmospheric transport. We use a nested version of GEOS-Chem as described by Kim et al. (2015) with $0.25° \times 0.3125°$ horizontal resolution over the North America window and adjacent oceans (9.75°-60°N, 130°-60°W), driven by GEOS-FP assimilated meteorological data from the NASA Global Modeling and Assimilation Office (GMAO). The same version of the GEOS-Chem has been applied to simulation of other chemical observations from the SEAC[4]RS campaign (Kim et al., 2015; Fisher et al., 2016; Marais et al., 2016; Travis et al.,

2016; Zhu et al., 2016; Yu et al., 2016; Chan Miller et al., 2017). The boundary conditions for the nested-grid simulation are from a $4° \times 5°$ global simulation by Turner et al. (2015) using methane emissions optimized with three years of GOSAT satellite data. The model uses a 3-D archive of monthly average OH concentrations from Park et al. (2004), with a lifetime of 8.9 years in the troposphere consistent with observational constraints (Prather et al., 2012; Turner et al., 2017). Loss by OH is irrelevant for our North American simulation since ventilation of the domain is much faster(Wecht et al., 2014). Since we

treat OH concentrations as decoupled from methane in the inversion, the relationship between emissions and concentrations is linear, so that $\mathbf{K}$ fully describes the GEOS-Chem model as applied to our problem.

The prior emission estimates for the inversion are taken from the $0.1° \times 0.1°$ gridded version of the EPA anthropogenic greenhouse gas emission inventory for 2012 (Maasakkers et al., 2016) and the mean wetland emissions from the $0.5° \times 0.5°$ monthly WetCHARTs extended ensemble for 2013 (Bloom et al., 2017). Figure 1 (top panels) shows the distribution of these



prior methane emissions over the inversion domain for August-September 2013. Emissions total 13.3 Tg a$^{-1}$ for anthropogenic sources and 13.0 Tg a$^{-1}$ for wetlands over these two months (expressed on an annual basis).

The SEAC$^4$RS DC-8 aircraft conducted 21 flights over the Southeast between August 6 and September 21, 2013. Methane was measured by gas chromatography from whole air flask samples and calibrated to the NOAA standard. Figure 2 (left panel)
shows the SEAC$^4$RS flight tracks and the spatial distribution of the methane measurements below 2 km altitude averaged over the $0.25° \times 0.3125°$ model grid. The mean observed vertical profile is shown in the right panel of Figure 2, and compared to the GEOS-Chem profile using the prior emissions. The model is unbiased in the free troposphere above 2 km, implying a successful representation of background methane by the boundary conditions. Model overestimation in the boundary layer below 2 km suggests that the prior US emissions are too high. In what follows we will use the SEAC$^4$RS observations over the
Southeast US below 2 km altitude for the inversion. This represents a total data set of $m = 652$ methane observations averaged over the $0.25° \times 0.3125°$ GEOS-Chem grid for individual flights.

We use the residual error method (Heald et al., 2004) to estimate the diagonal elements of the observational error covariance matrix $\mathbf{S_O}$. The method assumes that the mean bias between the observations and the model with prior emissions is to be corrected by the inversion, while the residual error represents the observational error including contributions from the instrument
and the transport model. Figure 3 shows the vertical profile of the residual error standard deviation (RSD) for the ensemble of the SEAC$^4$RS data over the Southeast US. The RSD is about 60 ppb below 2 km and 20 ppb in the free troposphere above. Subsetting the data by latitudinal bands gives similar results. We thus use 60 ppb for the standard deviation of the observational error (diagonal elements in $\mathbf{S_O}$). The instrument precision is better than 2 ppb (Simpson et al., 2002), thus most of that observational error is from the transport model (including representation error). We take $\mathbf{S_O}$ to be diagonal since error correlations
between boundary-layer observations on the GEOS-Chem grid are not significant (Wecht et al., 2014).

The inversion can in principle optimize emissions at the $0.25° \times 0.3125°$ grid resolution of the GEOS-Chem model, representing 3004 grid cells over the inversion domain. However, the aircraft observations do not have sufficient information to constrain emissions at that resolution. In order to reduce the dimensionality of the state vector, we project the 3004 grid cells onto 216 elements of a Gaussian mixture model (GMM) with radial basis functions based on spatial proximity and source type
patterns (Turner and Jacob, 2015). The use of the GMM allows us to retain high resolution of up to 25 km for major localized sources while degrading resolution in areas of weak or broadly distributed sources. Areas dominated by wetlands have resolution of 100-200 km in the GMM because they are broadly distributed. Individual state vector elements in the GMM have weighted influence functions over the $0.25° \times 0.3125°$ GEOS-Chem grid, so that the inversion effectively returns information on that fine grid even though the actual resolution of the inversion is coarser (Turner and Jacob, 2015).

The anthropogenic inventory of Maasakkers et al. (2016) and the wetlands inventory of Bloom et al. (2017) both include gridded error estimates that serve as the diagonal elements of the prior error covariance matrix $\mathbf{S_A}$. Maasakkers et al. (2016) found no significant spatial error correlation at $0.1° \times 0.1°$ resolution in their inventory while a variogram analysis across the elements of the WetCHARTS ensemble indicates a spatial error correlation length scale of 130 km. Therefore we take $\mathbf{S_A}$ to be diagonal.





## 3 Results and discussion

Figure 1 (bottom panels) shows the results of the inversion including the optimized posterior emissions, the corrections to the prior emissions, and the DOFS as measured by the diagonal elements of the averaging kernel matrix. Figure 4 (top panels) compare the observed boundary layer methane concentrations to the values simulated by GEOS-Chem with prior and posterior emissions (Figure 1). The simulation with prior emissions has a positive bias that is effectively corrected when using posterior emissions. The coefficient of determination ($R^2$) between model and observations increases from 0.30 to 0.50 when using posterior emissions. Figure 4 also evaluates the SEAC[4]RS inversion results with independent surface air observations from the three NOAA/ESRL surface network sites in the region (Andrews et al., 2014) and with methane column observations to from the TCCON site in Lamont, Oklahoma (Wunch et al., 2011; Wennberg et al., 2017). The posterior emissions improve the simulation of these independent data sets. GOSAT satellite observations are another source of independent data but the 2-month period is too sparse for useful evaluation (Wecht et al., 2014).

Total posterior emissions over the SEAC[4]RS domain are 15% (4 Tg a$^{-1}$) lower than the prior estimate (Figure 1). The inversion is able to constrain about 10 pieces of information in the spatial distribution of methane emissions as measured by the DOFS. It is strongly sensitive to the Gulf Coast and to large anthropogenic source areas such as the Floyd Shale in central Alabama. The posterior errors are 18%-30% over these regions. The scaling factors show large downward corrections of prior emissions in Louisiana and Mississippi, and along the Gulf Coast, where wetlands are the dominant sources. There are also downward corrections in southern West Virginia, where coal mines are dominant, and in the Haynesville Shale gas production region of northern Louisiana and southern Arkansas. On the other hand, there are significant upward corrections for the coal mines of southern Illinois and for the Floyd Shale in central Alabama.

We can attribute the $0.25° \times 0.3125°$ scaling factors from the inversion to specific methane source sectors by using the sector-resolved spatial patterns in the prior emission inventories, as described by Turner et al. (2015) but here with the improved anthropogenic source patterns from Maasakkers et al. (2016) and wetland emissions from (Bloom et al., 2017). Figure 5 compares our results with the prior emission totals for the different sectors in the Southeast US. We find a significant 27% (3.6 Tg a$^{-1}$) reduction in regional wetland emissions (mean of the WetCHARTs extended ensemble). By contrast, we find no significant regional bias in the EPA anthropogenic inventory for any of the major source sectors.

The WetCHARTs extended ensemble includes 18 wetland methane emission models intended to encompass the uncertainties in estimating wetland emissions (Bloom et al., 2017). The different models (ensemble members) use different datasets for wetland extent fraction $A$ [m$^2$ wetlands per m$^2$ surface area], heterotrophic respiration rate $R$ [mg C day$^{-1}$ per m$^2$ of wetland areas], temperature-dependent factor $q_{10}^{T/10}$ of C respired as CH4 [mg CH$_4$ per mg C] where $T$ is the surface skin temperature, and global scaling factors $s$. The wetland methane emission flux $E$ [mg CH$_4$ m$^{-2}$ day$^{-1}$] at a time $t$ and location $x$ for each of these members is given by

$$E(t,x) = sA(t,x)R(t,x)q_{10}^{T(x,t)}. \tag{5}$$

The 18-member ensemble consists of three temperature dependence factors ($q_{10} = 1, 2, 3$), three global scale factors ($s = 125, 166, 208$), and two wetland extent maps ($A$) from the Global Lakes and Wetlands Database (GLWD; Lehner and Dölla,





2004) and GLOBCOVER (Bontemps et al., 2011). The heterotrophic respiration rate ($R$) is the median output from the carbon data model framework (CARDAMOM; Bloom et al., 2016), and is not varied across that ensemble.

Figure 6 shows the Southeast US wetland emissions for each WetCHARTs member, along with the root-mean-square error (RMSE) of its spatial distribution relative to our optimized posterior estimate on the $0.25° \times 0.3125°$ grid. Consistency in
spatial distribution with our optimized estimate is indicated by a low RMSE. We find that the specification of wetland extent is the most systematic source of error in wetland emission estimates; all GLOBCOVER-based models underestimate wetland emissions, while all GLWD-based models overestimate emissions. Estimates using $q_{10} = 1$ (no temperature dependence in the CH4:C respiration ratio) exhibit the lowest RMSE values. The WetCHARTs ensemble mean used as prior for our inversion performs better than any individual member.

For anthropogenic emissions, Figure 5 shows that the inversion is consistent on the regional scale with the EPA sectoral inventory gridded by Maasakkers et al. (2016). Previous inversions using the EDGAR v4.2 inventory as prior found large underestimates over the Southcentral US that they attributed to a combination of oil/gas and livestock sources (Miller et al., 2013; Alexe et al., 2015; Turner et al., 2015). This is in contrast with our finding in particular for East Texas and Louisiana. The EDGAR v4.2 inventory has large errors in its source patterns (see Fig. 3 of Maasakkers et al. (2016)). It places almost all
oil/gas emissions in distribution centers instead of in production fields, and the resulting methane concentration underestimate over production fields (e.g., Floyd shale in East Texas) would be interpreted as a model error. These previous inverse studies also used low prior estimates of wetland emissions (2.7-5.9 Tg a$^{-1}$ for the contiguous US, 1.6-3.5 Tg a$^{-1}$ for the Southeast US, at the low end of the WetCHARTs ensemble in Figure 5), leading them to erroneously attribute methane underestimates to nearby anthropogenic sources.

Despite the good regional agreement of our inversion with the EPA (2016) inventory for the Southeast US for different sectors, there are large local biases that tend to cancel each other on a regional scale (e.g., Haynesville Shale vs. Floyd Shale for natural gas). This suggests that methane emission factors for the oil/gas sector are more variable than assumed in the EPA (2016) inventory.

## 4   Conclusions

We used extensive boundary layer methane observations from the SEAC$^4$RS aircraft campaign over the Southeast US in August-September 2013 to optimize methane emissions in that region with up to $0.25° \times 0.3125°$ spatial resolution and with detailed error characterization. The inversion used new state-of-science inventories as prior information, including the gridded version of the EPA (2016) national anthropogenic inventory from Maasakkers et al. (2016) and the WetCHARTs wetlands extended ensemble from Bloom et al. (2017). The inversion domain over the Southeast US accounts for 45% of national
methane emissions in the EPA inventory, and for 56% of wetland emissions over the contiguous US in the mean WetCHARTs estimate.

Our inversion results suggest that the EPA emission inventory has no significant bias on the regional scale for the major source sectors (livestock, oil/gas, waste, coal), while the mean of the WetCHARTs wetland ensemble needs to be reduced by





27% over the inversion domain. These results are supported by independent methane observations from the NOAA/ESRL surface network and from the TCCON site in Lamont, Oklahoma. The specification of wetland areal extent is the dominant source of error in the WetCHARTs ensemble. Results also indicate that a low temperature dependence for the CH4:C heterotrophic respiration ratio best explains the spatial variability of the posterior emissions. The mean of the WetCHARTs ensemble performs better than any individual ensemble member. Our finding of regional consistency with the EPA anthropogenic inventory is in contrast with previous inverse studies that found large underestimates. These inversions relied on EDGAR v4.2 anthropogenic source patterns that have large errors and also assumed low wetland emissions. Despite regional agreement we still find significant local discrepancies with the EPA inventory for the oil/gas sector, suggesting that methane emission factors are more variable than assumed in the inventory.

*Acknowledgements.* This work was funded by the NASA Earth Science Division. Part of this research was carried out at the Jet Propulsion Laboratory, California Institute of Technology, under a contract with NASA. This work was funded by a NASA Earth Sciences grant (#NNH14ZDA001N-CMS). Special thanks to D.R. Blake for providing SEAC$^4$RS aircraft methane observations (available at https://www-air.larc.nasa.gov/cgi-bin/ArcView/seac4rs#BLAKE.DONALD/). TCCON data were obtained from the TCCON Data Archive, hosted by CaltechData (http://tccondata.org). The NOAA data are available from the ObsPack portal (https://www.esrl.noaa.gov/gmd/ccgg/obspack/).



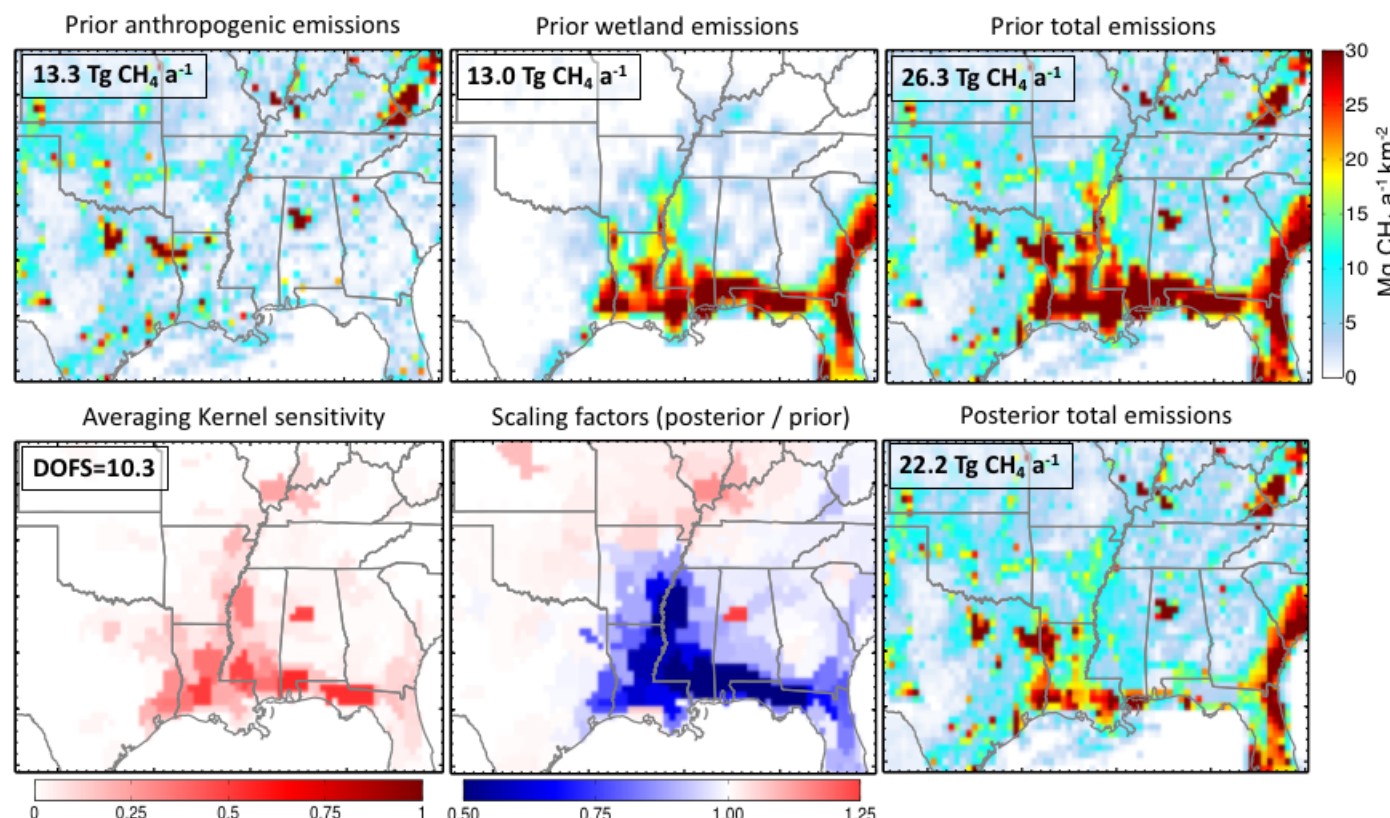

**Figure 1.** Methane emissions in the Southeast US in August-September 2013. The top panels show the prior anthropogenic and wetland methane emissions, and the bottom panels show the inversion results including posterior emissions, scaling factors (posterior/prior emission ratios), and the diagonal elements of the averaging kernel matrix for the inversion. The sum of these diagonal elements (trace of the averaging kernel matrix) quantifies the degrees of freedom for signal (DOFS) of the inversion. Annual emission rates for the SEAC[4]RS period are shown inset.



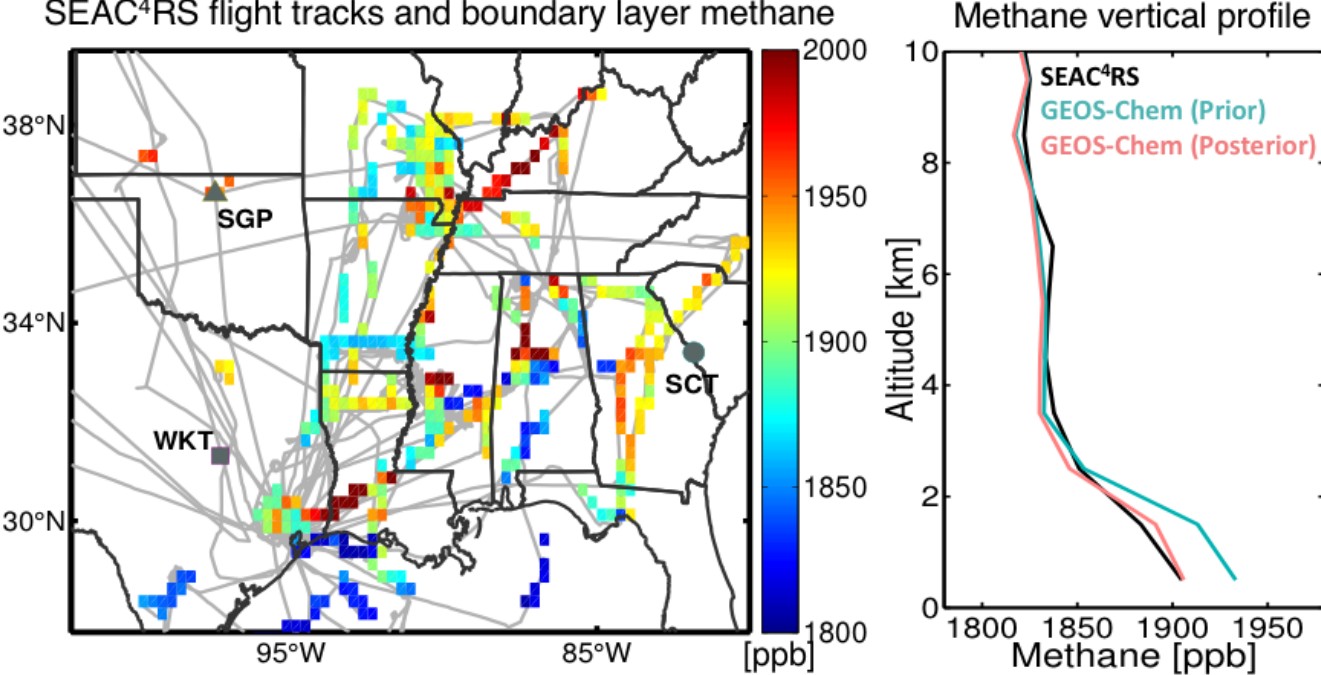

**Figure 2.** Boundary layer methane concentrations over the Southeast US measured during the SEAC[4]RS aircraft campaign (August 6-September 21, 2013). The left panel shows the flight tracks in grey and the methane measurements at 0-2 km altitude averaged over the $0.25° \times 0.3125°$ GEOS-Chem grid. The three NOAA/ESRL sites at SGP (Southern Great Plains, Oklahoma; $36.6°N$, $97.5°W$) , WKT (Moody, Texas; $31.3°N$, $97.3°W$), and SCT (Beech Island, South Carolina; $33.4°$, $81.8°W$) are indicated. SGP is co-located with the TCCON site at Lamont, Oklahoma. The right panel shows the mean methane vertical profiles over the Southeast US domain measured from the aircraft and simulated by GEOS-Chem using the prior and posterior emissions.





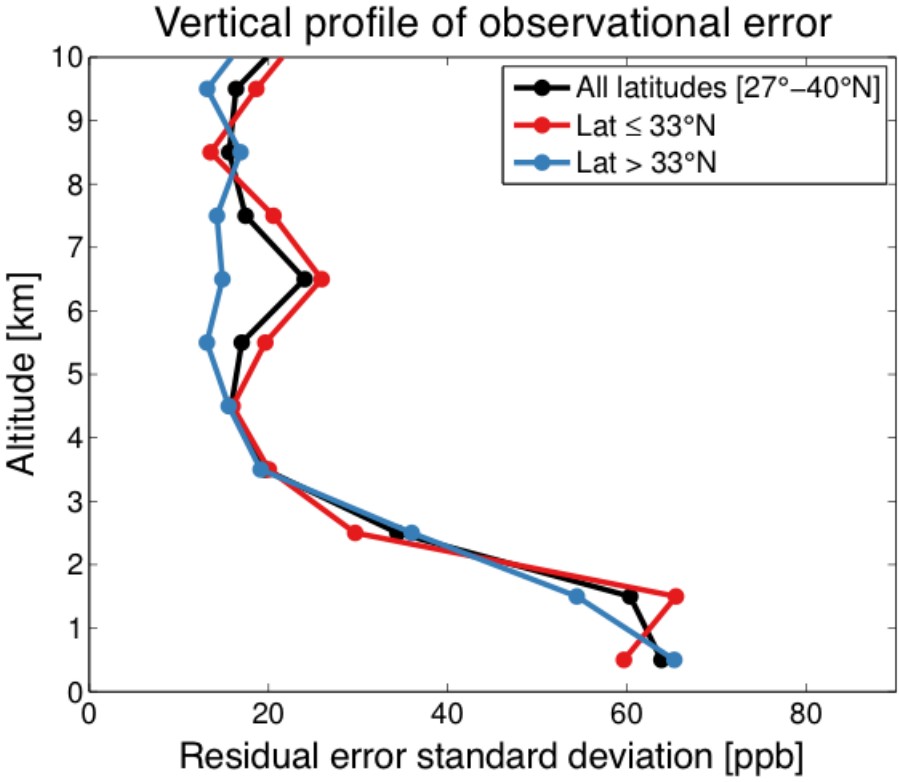

**Figure 3.** Residual standard deviations (RSDs) of the difference between SEAC[4]RS methane observations and the GEOS-Chem model with prior emissions, for 1-km altitude bins. These RSDs are used to define the observational error standard deviations for the inversion as described in the text. The observational error is mainly from GEOS-Chem (see text). Values are shown for two latitudinal ranges.



**Figure 4.** Evaluation of the SEAC$^4$RS inversion of methane emissions in the Southeast US for the August 6 - September 21, 2013 period. The top panels compare GEOS-Chem methane concentrations with the SEAC$^4$RS observations, using prior emissions (left) and posterior emissions (right). The middle panels compare GEOS-Chem methane concentrations with independent observations from the three NOAA/ESRL surface sites in the inversion domain (see Fig. 2 and caption). The bottom panels compare GEOS-Chem methane columns with TCCON hourly column observations at Lamont, Oklahoma (Wennberg et al., 2017), after correction for stratospheric bias in the model (Turner et al., 2015). The 1:1 lines (dashed) and the reduced-major-axis (black solid line) linear regressions are also shown, along with the coefficients of determination ($R^2$) and the slopes ($\pm 1\sigma$) derived from the bootstrap method.




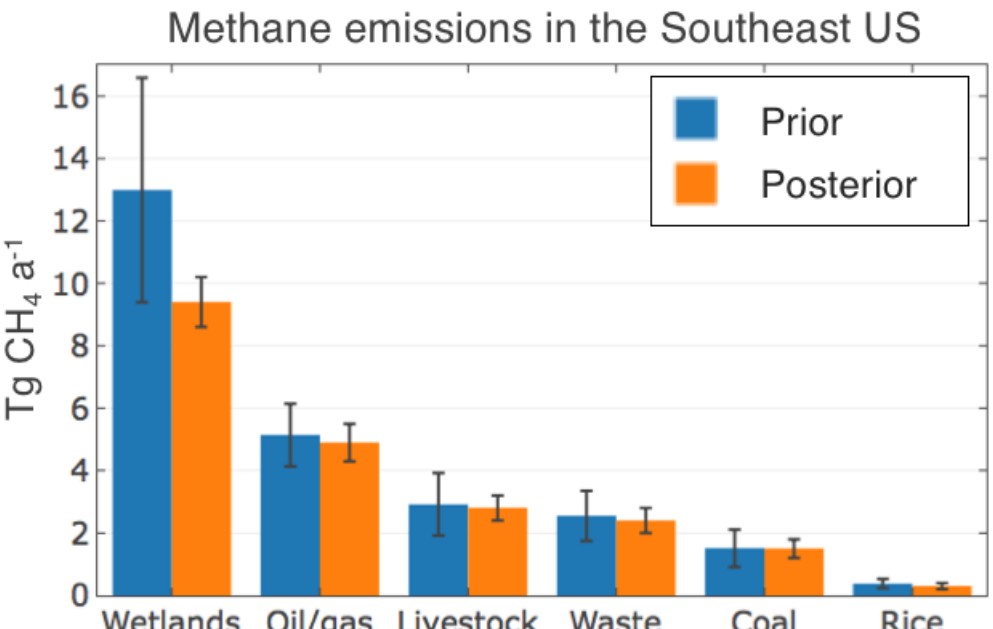

**Figure 5.** Prior and posterior methane emissions for the Southeast US domain of Fig. 1. The prior anthropogenic emissions are from the EPA national inventory for 2012 (EPA, 2016; Maasakkers et al., 2016) and the prior wetland emissions are the means of the WetCHARTs extended ensemble (Bloom et al., 2017). Error bars (one standard deviation) on sectoral emissions are from the prior and posterior error variances of our inversion .

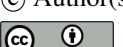



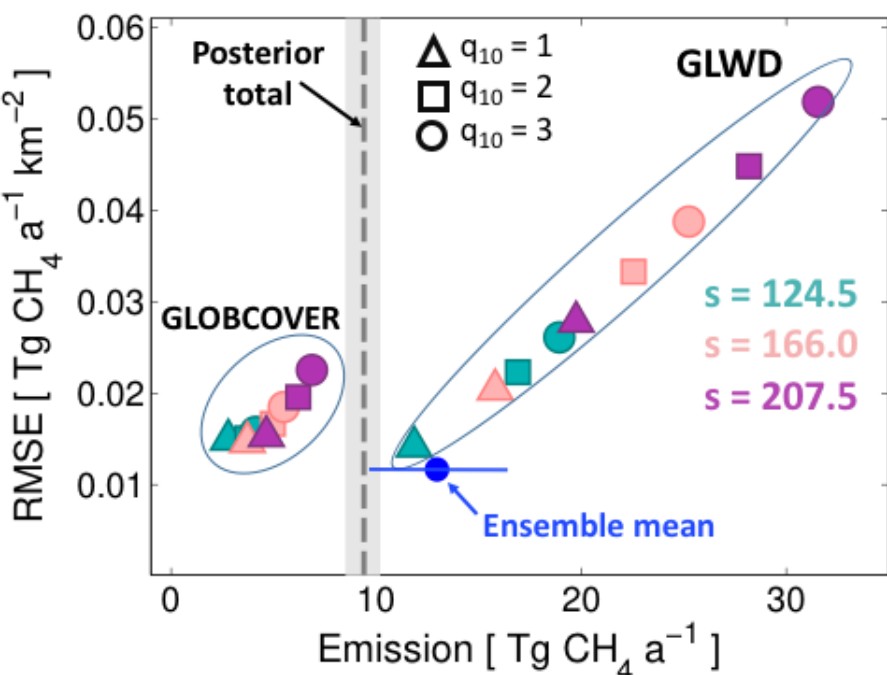

**Figure 6.** Range of wetland emission estimates for the Southeast US (domain of Figure 1). The figure shows the spread of the WetCHARTs extended ensemble and compares with the posterior emission estimate from our inversion in terms of emission total and root-mean-square error (RMSE) on the $0.25° \times 0.3125°$ spatial grid. WetCHARTs ensemble members use wetland areal extent data from either the GLOB-COVER (Bontemps et al., 2011) or GLWD (Lehner and Dölla, 2004) databases, as well as different estimates of temperature sensitivity $q_{10}$ and global scaling factors $s$ (see equation (5) and text). The posterior wetland emission estimate from our inversion is shown as dashed line with error standard deviation shaded. The mean of the WetCHARTs ensemble used as prior for our inversion is shown as blue solid circle with its associated error standard deviation.



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
