# Peer review of "High-resolution inversion of methane emissions in the Southeast US using SEAC4RS aircraft observations of atmospheric methane: anthropogenic and wetland sources"

_Atmospheric Chemistry and Physics, 2017_

## Referee Comment (RC1) · Anonymous Referee #1 · 19 Jan 2018

Summary/General comments: Sheng et al. present a high resolution inversion of SEAC4RS aircraft methane data to find optimal methane emissions in that region in that time frame. They find the new, gridded EPA inventory is consistent with their observations, while WETCHIMP methane emissions are found to be too high. This paper is well placed in ACP. Overall the paper is well written, clear, and adds to our understanding of methane emissions in the US. I have only a couple of concerns – once these are addressed I would encourage publication.

Major comments: I have two larger concerns: The definition of region: The region

studied is defined as the Southeast US and is illustrated in Figure 1. My problem is that the SEAC4RS data does not constrain emissions for this whole region. In fact, less than half the domain has a significant sensitivity (AK sensitivity) and thus is informed by the analysis. Important regions that have large fluxes in the prior inventories (fossil in TX, wetland in FL, GA, SC) are not constrained by the work but are included in the regional flux estimates. This should be corrected. At a minimum, the area defined should be compressed to not include large expanses with minimal surface sensitivity in the west. Even better, would be to use the AK sensitivity to filter only the domain where there is significant surface constraint. This could be illustrated in Figure 1, and that mask could be applied to the domain for flux estimation. This would be straightforward for the authors to do and would make the results more robust.

Transport error/60 ppb: I'm a little unclear on how 60 ppb was settled on as the observation-model error. Transport error could be significant, and I would like to see more discussion/explanation of how that is accounted for. If the 60 ppb is derived from the observations—I wonder if it is more representative of atmospheric variability?

Minor comments: Page 1 Line 3: The % will be lower once accounting for the above comment.

Page 1 Line 11: It's not clear you can state your work is in contrast with national scale work. Not only is the work on different scales, it is for different years, and, more importantly, different times of year. This study is focused on only Aug-Sept, whereas other studies have used 1+ year of data. This needs to be made clearer throughout that the results are for Aug.-Sept. only.

Page 2 Line 9: Should indicate here at least once if Tg C or Tg CH4.

Page 2 Lines 15-20: This is not strictly accurate. Miller et al., 2013 did not rely on EDGAR, as a geostatistical approach was used. I would suggest changes phrasing here to correct this. (A side comment—the Miller work did have little data in the Southeast so it was essentially unconstrained).

Page 4 Lines 5-6: I have some concerns about the way the flask data has been aggregated onto a grid here. It would be very helpful to see some continuous variables for the flights and where the flasks were sampled. This would enable the reader to assess if averaging the discrete flask samples is representative of 0.25 degree boxes, or if the samples are representative of smaller atmospheric features (such as plumes).

Page 4 Line 17: I'd like more on the 60 ppb.

Page 6 Lines 10-20. Care is needed here as mentioned before not all these studies relied to this level on EDGAR. Also, many of those studies were rather unsensitive to the SE, so they likely do not see the wetland emissions, as opposed to falsely attributing those emissions to anthropogenic sources. At the least, we cannot make the conclusion in lines 18-19.

Page 6 lines 29-31: these fractions will come down when accounting for region of sensitivity.

Page 6 line 7: Should specify this finding of regional consistency is for Aug.-Sept. only.

Figure 1: Zoom and add contours as described in major comment.

Figure 2: I struggle with the map figure as it is hard to interpret the methane concentration on a map like this where we don't know if it is when the mixing layer is deep or shallow or what the background value is on the given day. For example, are regions with blue and red adjacent indicative of high spatial variability or from different sampling?

Figure 5: Update when updating domain definition.

---

## Referee Comment (RC2) · Anonymous Referee #2 · 13 Feb 2018

**General comments**

This study uses observations of CH4 from aircraft campaigns in Aug-Sep 2013 to estimate CH4 emissions in the southeast US, a region with importance to the US total anthropogenic CH4 emission and with significant areas of wetlands. The authors use a Bayesian inversion method to estimate the emissions and uncertainties. The method is scientifically sound and the manuscript is well-written. However, there are a few outstanding points that should be clarified before publication. In particular, I think the results of the study would be strengthened by adding a sensitivity test to determine the
sensitivity of the results to the prior wetlands emission estimate used (see also specific comments). In summary, I recommend publication after minor revisions.

**Specific comments**

P1, L13: The authors state that the results of previous inversions finding higher emissions than EPA estimates were owing to too low estimates for wetland emissions. What estimate for the wetland emissions is the reference here?

P2, L19-20: While errors in the prior can bias the results, the extent to which this is a problem is dependent on the constraint from the observations and on the uncertainty assigned to the prior estimates, or in other words the degrees of freedom to adjust the prior. This statement should be qualified bearing in mind these other factors too.

P3, Eq.1: Strictly speaking, the RHS of this equation should be multiplied by 1/2

P3, L14: Here the authors say that inversions of methane are usually solved numerically, however, that is not the case. While global inversions of CH4 over many years usually use numerical adjoint methods, regional inversions (as in this study) also often use the analytical solution.

P3, L15: While it is true that the analytical method allows the posterior error covariance to be calculated directly, some numerical methods allow it to be estimated.

P3, L26: Did the global simulation optimized with GOSAT also include surface observations? This should be mentioned. Also, it has been shown that satellite-only optimizations can lead to be biases due to errors in the retrievals. Have comparisons of the CH4 mixing ratios from the optimized simulation against surface and/or aircraft observations been made?

P4, L12: I think it would be helpful if the authors would briefly explain the residual method, which is used to calculate the observation error variances.

P4, L28-29: I'm not sure how the inversion can return information at the fine scale
when it is performed at coarser scale. I think further explanation would be helpful here.

P5, L20-25: There is a little bit of overlap in the locations of the anthropogenic and wetland sources (see Fig. 1). Has this been factored into the posterior emission estimates per source type?

P6, L8-9: The WetCHARTs ensemble mean was used as the prior for the inversion, therefore, it is not surprising that the ensemble mean has the lowest RMSE compared to the posterior emissions, as the two estimates are not independent from one another. This should be stated. Also, have the authors looked at the Gain matrix to determine how well constrained the wetland emissions are by the observations? It would be valuable to test how sensitive the results are to using a different prior for the wetland emissions.

**Technical comments**

P1, L5: I don't think the term "state-of-science" exists (or at least I've never heard it before). I suggest replacing with "state-of-the-art" or "up-to-date" (also elsewhere in the manuscript).

**ACPD**

---

## Author Comment (AC1) · 29 Mar 2018

We have uploaded our response, updated manuscript (with the tracked changes) in the supplement.

Please also note the supplement to this comment:
https://www.atmos-chem-phys-discuss.net/acp-2017-1151/acp-2017-1151-AC1-supplement.zip

---

## Author Response (AR1)

We thank the referees for their helpful comments that have improved our paper.

**Anonymous Referee #1**

Summary/General comments: Sheng et al. present a high resolution inversion of SEAC4RS aircraft methane data to find optimal methane emissions in that region in that time frame. They find the new, gridded EPA inventory is consistent with their observations, while WETCHIMP methane emissions are found to be too high. This paper is well placed in ACP. Overall the paper is well written, clear, and adds to our understanding of methane emissions in the US. I have only a couple of concerns – once these are addressed I would encourage publication.

Major comments: I have two larger concerns: The definition of region: The region studied is defined as the Southeast US and is illustrated in Figure 1. My problem is that the SEAC4RS data does not constrain emissions for this whole region. In fact, less than half the domain has a significant sensitivity (AK sensitivity) and thus is informed by the analysis. Important regions that have large fluxes in the prior inventories (fossil in TX, wetland in FL, GA, SC) are not constrained by the work but are included in the regional flux estimates. This should be corrected. At a minimum, the area defined should be compressed to not include large expanses with minimal surface sensitivity in the west. Even better, would be to use the AK sensitivity to filter only the domain where there is significant surface constraint. This could be illustrated in Figure 1, and that mask could be applied to the domain for flux estimation. This would be straightforward for the authors to do and would make the results more robust.
We have updated Figure 1, Figure 5, and the text accordingly.

Transport error/60 ppb: I'm a little unclear on how 60 ppb was settled on as the observation-model error. Transport error could be significant, and I would like to see more discussion/explanation of how that is accounted for. If the 60 ppb is derived from the observations. I wonder if it is more representative of atmospheric variability?
60 ppb is derived from the statistics of residual errors (differences between observed and simulated values after removing the mean model bias). We have expanded discussion.

Minor comments: Page 1 Line 3: The % will be lower once accounting for the above comment.
We have updated the text as: "The Southeast US is a major source region for US methane emissions including large contributions from oil/gas production and wetlands."

Page 1 Line 11: It's not clear you can state your work is in contrast with national scale work. Not only is the work on different scales, it is for different years, and, more importantly, different times of year. This study is focused on only Aug-Sept, whereas other studies have used 1+ year of data. This needs to be made clearer throughout that the results are for Aug.-Sept. only.
Little regional or interannual variability is expected for anthropogenic emissions and so our results have general applicability. This is now stated in the text with references. For wetlands we have added a few statements that our results are for Aug-Sep 2013.

Page 2 Line 9: Should indicate here at least once if Tg C or Tg CH4.
Done.

Page 2 Lines 15-20: This is not strictly accurate. Miller et al., 2013 did not rely on EDGAR, as a geostatistical approach was used. I would suggest changes phrasing here to correct this. (A side comment: the Miller work did have little data in the Southeast so it was essentially unconstrained).

We corrected this.

Page 4 Lines 5-6: I have some concerns about the way the flask data has been aggregated onto a grid here. It would be very helpful to see some continuous variables for the flights and where the flasks were sampled. This would enable the reader to assess if averaging the discrete flask samples is representative of 0.25 degree boxes, or if the samples are representative of smaller atmospheric features (such as plumes).

We now plot the continuous data in Fig. 2, and update the text accordingly.

Page 4 Line 17: I'd like more on the 60 ppb.

See the response in major comments.

Page 6 Lines 10-20. Care is needed here as mentioned before not all these studies relied to this level on EDGAR. Also, many of those studies were rather unsensitive to the SE, so they likely do not see the wetland emissions, as opposed to falsely attributing those emissions to anthropogenic sources. At the least, we cannot make the conclusion in lines 18-19.

Agree. We now omit this conclusion.

Page 6 lines 29-31: these fractions will come down when accounting for region of sensitivity.

Done.

Page 6 line 7: Should specify this finding of regional consistency is for Aug.-Sept. only.

See response above.

Figure 1: Zoom and add contours as described in major comment.

We have updated Fig. 1.

Figure 2: I struggle with the map figure as it is hard to interpret the methane concentration on a map like this where we don't know if it is when the mixing layer is deep or shallow or what the background value is on the given day. For example, are regions with blue and red adjacent indicative of high spatial variability or from different sampling?

See the response for "Page 4 Lines 5-6:…"

Figure 5: Update when updating domain definition.

Done.

**Anonymous Referee #2**

General comments
This study uses observations of CH4 from aircraft campaigns in Aug-Sep 2013 to estimate
CH4 emissions in the southeast US, a region with importance to the US total
anthropogenic CH4 emission and with significant areas of wetlands. The authors use
a Bayesian inversion method to estimate the emissions and uncertainties. The method
is scientifically sound and the manuscript is well-written. However, there are a few
outstanding points that should be clarified before publication. In particular, I think the
results of the study would be strengthened by adding a sensitivity test to determine the
sensitivity of the results to the prior wetlands emission estimate used (see also specific
comments). In summary, I recommend publication after minor revisions.
See the response below in specific comment.

Specific comments
P1, L13: The authors state that the results of previous inversions finding higher emissions
than EPA estimates were owing to too low estimates for wetland emissions. What
estimate for the wetland emissions is the reference here?
We have removed this statement (see response to Referee #1). It's not relevant now.

P2, L19-20: While errors in the prior can bias the results, the extent to which this is a
problem is dependent on the constraint from the observations and on the uncertainty
assigned to the prior estimates, or in other words the degrees of freedom to adjust the
prior. This statement should be qualified bearing in mind these other factors too.
We now add: "… this depends on the constraint from the observations and on the uncertainty
assigned to the prior estimates."

P3, Eq.1: Strictly speaking, the RHS of this equation should be multiplied by ½
We have corrected the equation.

P3, L14: Here the authors say that inversions of methane are usually solved numerically,
however, that is not the case. While global inversions of CH4 over many years
usually use numerical adjoint methods, regional inversions (as in this study) also often
use the analytical solution.
We have updated the text as:
"…analytically or numerically using an adjoint method (Jacob et al., 2016). Unlike adjoint-based
inversions, analytical solution provides direct error characterization …".

P3, L15: While it is true that the analytical method allows the posterior error covariance

to be calculated directly, some numerical methods allow it to be estimated.
We have added 'direct'

P3, L26: Did the global simulation optimized with GOSAT also include surface observations? This should be mentioned. Also, it has been shown that satellite-only optimizations can lead to be biases due to errors in the retrievals. Have comparisons of the CH4 mixing ratios from the optimized simulation against surface and/or aircraft observations been made?
The GOSAT optimized emissions are evaluated with independent surface observations. We now mention this in the text.

P4, L12: I think it would be helpful if the authors would briefly explain the residual method, which is used to calculate the observation error variances.
We now do so. See response to Referee #1.

P4, L28-29: I'm not sure how the inversion can return information at the fine scale when it is performed at coarser scale. I think further explanation would be helpful here.
The statement is not strictly accurate. We have removed it.

P5, L20-25: There is a little bit of overlap in the locations of the anthropogenic and wetland sources (see Fig. 1). Has this been factored into the posterior emission estimates per source type?
The sources are in fact well separated on the $0.25^o \times 0.3125^o$ grid and we now say so.

P6, L8-9: The WetCHARTs ensemble mean was used as the prior for the inversion, therefore, it is not surprising that the ensemble mean has the lowest RMSE compared to the posterior emissions, as the two estimates are not independent from one another. This should be stated. Also, have the authors looked at the Gain matrix to determine how well constrained the wetland emissions are by the observations? It would be valuable to test how sensitive the results are to using a different prior for the wetland emissions.
We agree and now state this.
Results will be sensitive to different wetland ensemble members in particular those smallest or largest members, assuming the same prior uncertainty. We now also state this in the text.

Technical comments
P1, L5: I don't think the term "state-of-science" exists (or at least I've never heard it before). I suggest replacing with "state-of-the-art" or "up-to-date" (also elsewhere in the manuscript).
Done.

[revised manuscript text omitted]

**Figure 1.** Methane emissions in the Southeast US in August-September 2013. The top panels show the prior anthropogenic and wetland methane emissions, and the bottom panels show the inversion results including posterior emissions, scaling factors (posterior/prior emission ratios), and the diagonal elements of the averaging kernel matrix  representing the sensitivity of the inversion results to the observations. The sum of these diagonal elements over the domain (trace of the averaging kernel matrix) quantifies the degrees of freedom for signal (DOFS) of the inversion.  Numbers inset in the emission panels are the regional totals expressed as annual means for clarity (i.e., assuming that August-September emission rates hold for the  rest of the year). Values in parentheses are  the totals for the region with averaging kernel sensitivities larger than 0.05 (stippled areas in lower left panel).

[Figure]

**Figure 2.** Boundary layer methane concentrations over the Southeast US measured during the SEAC[4]RS aircraft campaign (August  6-September 21, 2013). The left panel shows the flight tracks in grey and the methane measurements at 0-2 km altitude. The three NOAA/ESRL sites at SGP (Southern Great Plains, Oklahoma; 36.6°N, 97.5°W) , WKT (Moody, Texas; 31.3°N, 97.3°W), and SCT (Beech Island, South Carolina; 33.4°, 81.8°W) are indicated. SGP is co-located with the TCCON site at Lamont, Oklahoma. The right panel shows the mean methane vertical profiles over the Southeast US domain measured from the aircraft and simulated by GEOS-Chem using the prior and posterior emissions.

[Figure]

**Figure 3.** Residual standard  deviation (RSD) of the difference between SEAC[4]RS methane observations and the GEOS-Chem model with prior emissions, for 1-km altitude bins.  The RSD defines the observational error standard deviations for the inversion as described in the text.  Values are shown for two latitudinal bands.

[Figure]

**Figure 4.** Evaluation of the SEAC[4]RS inversion of methane emissions in the Southeast US for the August 6 - September 21, 2013 period. The top panels compare GEOS-Chem methane concentrations with the SEAC[4]RS observations, using prior emissions (left) and posterior emissions (right). The middle panels compare GEOS-Chem methane concentrations with independent observations from the three NOAA/ESRL surface sites in the inversion domain (see Fig. 2 and caption). The bottom panels compare GEOS-Chem methane columns with TCCON hourly column observations at Lamont, Oklahoma (Wennberg et al., 2017), after correction for stratospheric bias in the model (Turner et al., 2015). The 1:1 lines (dashed) and the reduced-major-axis (black solid line) linear regressions are also shown, along with the coefficients of determination ($R^2$) and the slopes ($\pm 1\sigma$) derived from the bootstrap method.

[Figure]

**Figure 5.** Prior and posterior methane emissions  in the Southeast US (domain of Fig.  1) for August-September 2013. The prior anthropogenic emissions are from the EPA national inventory for 2012 (EPA, 2016; Maasakkers et al., 2016) and the prior wetland emissions are the means of the WetCHARTs extended ensemble (Bloom et al., 2017). Error bars (one standard deviation) on sectoral emissions are from the prior and posterior error variances of our inversion. Methane emissions in the subdomain with averaging kernel sensitivities larger than 0.05 (Figure 1) are also indicated.

[Figure]

**Figure 6.** Range of wetland emission estimates  in the Southeast US (domain of Figure 1)  for August-September 2013. 
[revised manuscript text omitted]